# Variability of External Intensity Comparisons between Official and Friendly Soccer Matches in Professional Male Players

**DOI:** 10.3390/healthcare9121708

**Published:** 2021-12-08

**Authors:** Hadi Nobari, João Paulo Brito, Jorge Pérez-Gómez, Rafael Oliveira

**Affiliations:** 1Department of Physical Education and Sports, University of Granada, 18010 Granada, Spain; 2Department of Physiology, School of Sport Sciences, University of Extremadura, 10003 Cáceres, Spain; 3Sports Scientist, Sepahan Football Club, Isfahan 81887-78473, Iran; 4Department of Exercise Physiology, Faculty of Educational Sciences and Psychology, University of Mohaghegh Ardabili, Ardabil 56199-11367, Iran; 5Sports Science School of Rio Maior, Polytechnic Institute of Santarém, 2140-413 Rio Maior, Portugal; jbrito@esdrm.ipsantarem.pt; 6Research Center in Sport Sciences, Health Sciences and Human Development, 5001-801 Vila Real, Portugal; 7Life Quality Research Centre, 2140-413 Rio Maior, Portugal; 8HEME Research Group, Faculty of Sport Sciences, University of Extremadura, 10003 Cáceres, Spain; jorgepg100@gmail.com

**Keywords:** performance, load monitoring, high-speed running, match load, player load, sprint

## Abstract

The aims of this study were to compare the external intensity between official (OMs) and friendly matches (FMs), and between first and second halves in the Iranian Premier League. Twelve players participated in this study (age, 28.6 ± 2.7 years; height, 182.1 ± 8.6 cm; body mass, 75.3 ± 8.2 kg). External intensity was measured by total duration, total distance, average speed, high-speed running distance, sprint distance, maximal speed and body load. In general, there was higher intensity in OMs compared with FMs for all variables. The first half showed higher intensities than the second half, regardless of the type of the match. Specifically, OMs showed higher values for total sprint distance (*p* = 0.012, ES = 0.59) and maximal speed (*p* < 0.001, ES = 0.27) but lower value for body load (*p* = 0.038, ES = −0.42) compared to FMs. The first half of FMs only showed lower value for body load (*p* = 0.004, ES = −0.38) than FMs, while in the second half of OMs, only total distance showed a higher value than FMs (*p* = 0.013, ES = 0.96). OMs showed higher demands of high intensity, questioning the original assumption of FMs demands. Depending on the period of the season that FMs are applied, coaches may consider requesting higher demands from their teams.

## 1. Introduction

An alert was recently launched for the soccer sports community about the need to discuss the global phenomenon of pre-season soccer matches as friendly matches (FMs) [1]. Pre-season is a specific period where the main objective is the acquisition of individual and collective adaptations that allow starting the competition adequately [2]. Usually, pre-season lasts four to six weeks and is of critical importance to develop high-level performance in soccer [3], which is supposed to be conducted with the aim of maximizing players’ participation in team training sessions to develop technical, psychological, physical and tactical performance [4,5]. From a conditioning point of view, the pre-season is characterized by a high volume of training and a gradually increasing intensity [1,6]; however, the improvement of strategical and tactical training should also be a major aim in this period [7].

According to the requirements of the field position, to reach a high level of physical performance at the elite competitive level, the analysis of match demands is the only feasible way to establish physical conditioning standards to be implemented in players. Some studies analysed the demand patterns during elite-level soccer match play [4,8,9]. For instance, they found that, during competitive matches, several intermittent periods of high-intensity activity such as high-speed running distance (HSRD) interspersed with low intensity periods occurred. Despite being a common practice, to use small-sided games during the pre-season in professional and semi-professional soccer, they do not reflect the intensity of the competitive game, and therefore coaches make use of FMs [8,9,10] in an attempt to improve physical fitness and skill development of the team [8]. This has been one of the aspects that have led to a greater use of FMs for the squads’ preparation [1,8].

The vast majority of FMs take place in the pre-season, where commercial imperatives are increasingly present in the team’s performance during this period, which can condition the training intensity and the pressure placed on players [1,11,12]. Consequently, these matches have been scheduled increasingly closer to the beginning of the pre-season, causing an extra physical and psychological demand to the players and coaching staff. This has had a major impact on strength and conditioning processes, which need to be much faster, which therefore require skipping important phases in the training process to apply higher intensity values than expected for the period [1,11]. It will probably generate unwanted consequences at different levels. Moreover, the “need to win” all matches contributed as an additional stressing factor [1,12]. In addition, an inappropriate training in the pre-season can be associated with a higher number of injuries through the in-season [13]. Beyond the training process, the higher number of matches played with a similar intensity to the official matches [14,15] may be associated with a higher injury rate [16]. It had been shown that high levels of training and matches are associated with higher risk of illness and injury [17,18,19,20]. 

Some studies reported that the pre-season is the period with the highest training intensity [1,21]. For instance, Jeong et al. [21] showed higher values of mean heart rate at 124 ± 7 beats/min and session-rated perceived exertion at 4343 ± 329 arbitrary units (AU) in pre-season, while lower values were revealed (heart rate, 112± 7 beats/min; session-rated perceived exertion 1703 ± 173 AU) during in-season [21]. 

The pre-season is currently considered by most teams as the period with the highest intensity and a high risk of non-traumatic injuries [1,6,22]. Conversely, Coppalle et al. [18] reported that this period is not associated with performance of the team because there are many more factors such as technical and tactical levels, opponents and environment that can influence the performance. Nonetheless and due to the commercial commitments, the pressure to win all matches, which also includes FMs, has increased. In this sense and according to Calleja-Gonzalez et al. [1], it was suggested to analyse external measures between official matches (OMs) and FMs. 

Currently, tracking systems are used in professional clubs for better external intensity management [22]. The most used technologies are micro-sensors devices, usually known as global positioning systems (GPS) [23,24,25]. They have shown to provide reliable and valid measures of the physical activity profile of team sports [26,27,28]. They allow to quantify several running distance speeds and accelerometery-based measures which are associated with the physical demands performed (e.g., training or matches). Usually, the external monitoring is known as external load [29,30,31]. However, due to the misuse of the load concept, “intensity” is used instead of “load” [29]. 

The development of high levels of physical performance is essential for performance in soccer. For that reason, it is necessary to develop a specific range of physical qualities [26] in order to best each the physical and physiological demands for the matches [8]. In this sense, some investigations have analysed the differences between the physical demands of training sessions or small-sided games in OMs [8,30] or FMs [11,12,32].

Another determinant aspect in modern soccer is the ability of players to maintain high levels of intensity during both halves of the match, and although some studies reported a decline in the total distances covered and HSRD in the second half compared to the first half [31,32], some studies in soccer and other sports have reported no differences between halves [4,33]. These fluctuations in intensity running can be caused by a variety of factors, including tactical alterations, the quality of the match, or the players’ level of preparation [31]. 

Based on previous literature, our working hypothesis was to verify if there is a decrement on players’ performance toward the end of match and between first and second halves for OMs and FMs. Understanding soccer players’ match-related demands and fatigue profiles likely helps with developing conditioning programs that increase team performance [1,34]. Despite match physical demands being a frequent study topic during the last years, the novelty of this research relies on the comparison for the determination of physical FMs and OMs demands. Therefore, the aims of this study were: to compare the running distances variables and body load between OMs and FMs; to compare all variables between first and second halves for OMs and FMs, respectively. It was hypothesised that some GPS measures would present higher values in FMs than in OMs, and that the first half of the matches would display higher intensity levels than the second half. 

## 2. Materials and Methods

### 2.1. Design 

A cohort study was conducted to identify differences between FMs and OMs during the season through GPS-derived variables. This professional soccer team had participated in the highest level of the Iranian Premier League called the Persian Gulf. In this league, teams were allowed to use GPS in competitions. All external monitoring and receiving information were performed by GPS with model GPSPORTS systems Pty Ltd., and SPI High-Performance Unit (HPU), Canberra, Australia. Finally, for the present study, 27 OMs and 10 FMs were analysed. The characteristics of the weeks and matches are presented in Table 1.

### 2.2. Participants

Twelve professional players were selected according to the inclusion criteria (age, 28.6 ± 2.7 years; height, 182.1 ±8.6 cm; body mass, 75.3 ± 8.2 kg; body mass index, 22.6 ±0.7 kg/m^2^) and consisted of participating in at least three consecutive matches and training with the team at least three training sessions a week. The anthropometric measurements were performed by specialists at the Iran Football Medical Assessment and Rehabilitation Center (http://ifmarc.ir/) (8 August 2021). In order to measure height and weight, the participants stood without shoes and with only shorts. For both measurements, a portable stadiometer (accuracy of ±5 mm) and balance weighting scales (accuracy of ±0.1 kg) (Seca model 207, Germany) was used. Body mass index (kg/m^2^) was calculated through the formula: weight/height^2^. The exclusion criteria of this study were players who did not attend training for more than two weeks, who were excluded from the study for any reason. In addition, the goalkeepers were omitted from the study. After coordination and obtaining official permission from the club director, an introductory session with the players as well as the team staff was conducted for the experimental approach of this study and individual consent was obtained from the players. This study was approved by the ethics committee of the University of Isfahan and Mohaghegh Ardabili University. During the study, the Helsinki Declaration was also considered for human studies.

### 2.3. Monitoring External Measures

GPS receiver specifications. During the season, all workouts and match sessions were monitored using GPSPORTS systems Pty Ltd. (Model: SPI High Performance Unit, Canberra, Australia) for professional athletes, which includes a 15 Hz position GPS and a tri-axial accelerometer to collect body load data. According to a previous study, this device has a high validity and inter-unit reliability within ±2% based on root mean square error [35]. There were no reported adverse weather conditions to affect data collection. 

Prior to the start of the match, belts were worn, and after the cooldown session post-match, they were collected. Then, GPS was placed in the dock system that allow downloading the information to save it through the Team AMS software. The same steps were described in a previous study [36]. 

According to the aims of the present study, match duration, total distance, average speed, HSRD (18–23 km·h^−1^), total sprint distance (>23 km·h^−1^), maximal speed (MS) and body load were measured.

### 2.4. Statistical Analysis

SPSS version 22.0 (SPSS Inc., Chicago, IL, USA) was used to analyse GPS data. First, the participants and GPS measures were described through descriptive statistics. Second, to verify the assumption normality and homoscedasticity of the several measures, Shapiro-Wilk and Levene’s tests were applied, respectively. 

In order to accomplish the study aims of comparing OMs vs. FMs and 1st vs. 2nd halves, *T* tests with 95% confidence interval (CI) were conducted. A *p* ≤ 0.05 was considered for statistical significance. In addition, *t* test family sample power was calculated for a post hoc compute achieve power (α level = 0.05, effect size = 0.8 and n = 12) by the G-Power [37]. There was an actual power of 83% for the present analysis and sample.

The last step consisted of the effect size (ES) calculation with CI (95%) to determine the magnitude of effects which was then analysed considering the range intervals: <0.2 = trivial, 0.2 > 0.6 = small effect, 0.6 > 1.2 = moderate effect, 1.2 > 2.0 = large effect and >2.0 = very large [38]. 

## 3. Results

Table 2 presents descriptive results, comparisons between first and second halves and full data between OMs and FMs. 

The comparisons between first half from OMs vs. FMs showed no significant differences in duration, but HSRD, sprint distance, and MS showed higher values in OMs, while body load presented higher values in FMs (all, *p* < 0.05, small effect size, which means a low power in considering the statistical power). 

The comparisons between the second half from OMs vs. FMs showed higher values for duration and total distance (all, *p* < 0.05, moderate effect size), but the other variables did not present differences. 

Considering full-match comparisons, total sprint distance and MS showed higher values in OMs than FMs, while body load showed higher values in FMs than OMs (all, *p* < 0.05 with small effect size, which mean a low statistical power). The other variables did not present significant differences. 

Comparisons between first and second halves for OMs and FMs, respectively, are presented in Table 3. Regarding OMs, there were higher duration, total distance and body load in the first half of OMs (all, *p* < 0.05, moderate to large effect size). Regarding FMs, there were higher duration, total distance and body load in the first half of FMs (all, *p* < 0.05, moderate effect size).

## 4. Discussion

The study’ aims were: to compare the running distances variables and body load between OMs and FMs; to compare all variables between first and second halves for OMs and FMs, respectively. The impact of the current phenomenon of the high-level of FMs which is related to its commercial imperatives and the additional stressing factor of the “need to win” [12] matches was the rationale for the present study. The need to win matches implies to understand soccer players’ match-related demands and fatigue profiles that will likely help in developing conditioning programs to increase team performance, wellness and to reduce injuries, illnesses [34]. 

In general, the results indicated some differences between OMs and FMs throughout the season; however, in the full-match data, there was no difference in OMs and FMs in duration, total distance, average speed and HSRD. The major findings were found in sprint distance and MS, where higher values were found in OMs, while body load showed higher values in FMs. According to Akenhead et al. [39] and Ade et al. [40] the external intensity imposed by OMs is often high because of the large amount of high-intensity activity demands required, such as accelerations and high-speed running. However, in the present study, despite the sprint distance and the maximal speed being higher in the OMs, the body load values are higher in the FMs, which can eventually be understood due to the coach’s tactical options, such as a more tactical positioning of the team and in the greater pressure to regain ball’ possession, due to the “need not to lose” the game. Thus, a greater number of impact/tackle/collision actions for ball recovery can cause a higher BL [41]. 

Usually, body load was used [41,42] to access the physiological demands of different sports. However, Gomez-Piriz et al. [43] analysed body load validity through the analysis of relationship with session-rated perceived exertion during training session and reported that the relation between session-rated perceived exertion and body load was weak and non-linear despite being significant. For this reason, it was suggested that body load may not be a valid measure to assess intensity in soccer. Since it uses an algorithm that calculates the total measure body load, it could be limited on this ability to globally quantify soccer-specific intensity. This measure also known as player load is not rigorous in identifying the influence that mode of motion and ball actions have on the energy expenditure during different actions in soccer [43]. Thus, Reilly and Bowen [44] reported that modes of displacement, such as running backward, running sideways and changing direction, can accentuate the metabolic charge. The combination of accelerometery, magnetometry and GPS software with match recordings may provide more insight into categorization of forces/accelerations received/exerted during the many contact elements within the game.

Some studies [10,45] recognized that accumulated measures of accelerometery can provide a different construct of the training process versus internal physiological intensity (i.e., rated perceived exertion, heart rate and blood lactate concentration). Nonetheless, they constitute valid measures to quantify the physical demands of the players [41].

Regarding the analysis of the lowest total distance observed in the FMs, there may possibly be a consequence of the fact that in these matches sometimes the number of interruptions is higher for information/substitutions.

Another factor that may explain the obtained results is that during the in-season, coaches tended to reduce training intensity [46] to allow the players to recover and reach the match at optimum fitness levels. 

However, pre-season is characterized by having a high weekly intensity, both due to the intensity assigned in the training sessions and the number of FMs, which can eventually condition fatigue. This possibly reflects the low priority of the coaches to prepare their teams before FMs during pre-season and can consequently contribute to higher accumulated fatigue level [8]. Campos-Vázquez et al. [8] reinforced the finding of FMs being the session with highest intensity during pre-season compared with data from training. Such results highlight the importance of playing FMs during the pre-season and/or to compensate the OMs absence during some periods of the in-season. Nevertheless, further investigations should aim to clarify if FMs actually reproduce the demands of OMs.

When comparing the first and second halves between OMs and FMs, higher values were found in the first half of OMs in the HSDR, total sprint distance and maximal speed variables. Once again, body load is higher in FMs. It can be inferred that the decisive actions of the game that usually underlie high intensity activities occur with greater magnitude in the first half of the OMs. In the second half, there were lower values of duration and total distance covered in FMs.

In the present study, a decrease of total distance was observed in the second half in both type of matches. However, there seemed to be no change in the intensity of the matches, although Mortimer et al. [47] reported that accumulated fatigue may contribute to a reduction in match intensity during the second half of a soccer match [48]. However, the reduction of total distance could be associated with the progressive use of glycogen during the game, which decreases performance in the second half [49].

As a conclusion of the analysis of our results, it is verified that the OMs present higher values in the maximal speed and in total sprint distance, which indicates that the intensity is higher in the displacements that underlie the decisive actions of the game.

Regarding the concern recently expressed by Calleja-Gonzalez et al. [1] that FMs are becoming less and less friendly, we found that in all friendly games performed, the intensity was lower than OMs. 

The present study points out, as a main limitation, the small sample size. Only players with at least three consecutive matches participated, which did not allow to provide further insights into the players with lower match participation, known in other studies as non-starters. Furthermore, because all players were from the same team, it is unclear whether the results obtained would be generalizable to other teams and competitive levels. Nevertheless, the present study represents the actual training and competition environment from athletes. 

## 5. Conclusions

The results of this study provided evidence for the difference in activity patterns between OMs and FMs in male professional soccer players. Specifically, OMs showed higher demands in the high-intensity domain, questioning the original assumption of FMs demands.

Furthermore, and because of this study, the use of FMs within the pre-season phase or during the in-season should warrant additional care when planned between high-intensity and high-volume training. For instance, pre-season FMs should be prepared to progressively increase the intensity in training program, which theoretically means that lower intensity should be applied in an early phase of the pre-season period when compared to the final phase of in-season periods. In addition, during in-season, FMs should be performed in the weeks without OMs in order to keep a day in the week with higher intensity, once matches constitute the most demanding intensity to players.

Despite all findings from this study, the results should be carefully interpreted due to the small sample size. Therefore, it is suggested to conduct more studies with identical design to confirm the present findings. 

## Figures and Tables

**Table 1 healthcare-09-01708-t001:** Characterization of the weeks and matches included for analysis.

Weeks	Type of Matches
1–2	Not included
3–5	Friendly
6–11	Official
12	Not included
13–16	Official
17	Friendly
18–21	Official
22	Non included
23–24	Official
25	Not included
26	Official
27	Not included
28	Friendly
29	Not included
30–31	Friendly
32–33	Official
34	Friendly
35–37	Official
38	Friendly
39	Official
40	Not included *
41–44	Official
45	Not included
46	Not included *
47	Friendly
48	Official

* weeks with two official matches.

**Table 2 healthcare-09-01708-t002:** Comparison of full match-day, 1st and 2nd halves between official matches and friendly matches, mean ± standard deviation and CI (95%).

**Full-Match**	**Official Matches**	**Friendly Matches**	** *p* **	**CI (95%)**	**Effect Size**
Duration (min)	87.9 ± 11.6 (80.5–95.3)	85.8 ± 4.1 (83.2–88.4)	0.514	−4.7, 8.9	0.24 (−57, 1.04)
Total Distance (m)	9424.7 ± 1224.5 (8646.7–10,202.7)	9125.7 ± 1224.5 (8697.8–9553.5)	0.435	−514.0, 1112.1	0.24 (−57, 1.04)
Average speed (m/min)	107.9 ± 11.4 (100.6–115.2)	106.8 ± 11.4 (99.5–114.0)	0.585	−3.4, 5.7	0.10 (−0.71, 0.89
HSRD (m)	241.7 ± 82.2 (189.4–293.9)	220.5 ± 74.7 (173.1–268.0)	0.274	−19.3, 61.5	0.27 (−0.54, 1.06)
Total sprint distance (m)	28.4 ± 9.5 (22.3–34.4)	21.8 ± 12.7 (13.7–29.9)	0.012 *	1.8, 11.3	0.59 (−0.25, 1.38)
Maximal speed (km·h^−1^)	29.0 ± 1.2 (28.2–29.8)	28.7 ± 1.0 (28.0–29.3)	<0.001 *	−0.1, 0.8	0.27 (−0.54, 1.07)
Body Load (au)	157.5 ± 38.9 (132.9–182.2)	179.8 ± 63.5 (139.5–220.2)	0.038 *	−43.1, −1.5	−0.42 (−1.22, 0.40)
**1st Half**	**Official Matches**	**Friendly Matches**	** *p* **	**CI (95%)**	**Effect Size**
Duration (min)	47.1 ± 1.8 (45.9–48.2)	47.2 ± 2.7 (45.5–48.9)	0.872	−1.9, 1.6	−4.40 (−5.69, −2.81)
Total Distance (m)	5181.5 ± 412.7 (4919.2–5443.7)	5123.1 ± 375.5 (4884.5–5361.7)	0.494	−123.3, 240.0	0.15 (−0.66, 0.94)
Average speed (m/min)	110.2 ± 9.5 (104.2–116.3)	108.9 ± 10.0 (102.5–115.2)	0.595	−4.1, 6.8	0.13 (−0.67, 0.93)
HSRD (m),	137.7 ± 58.9 (100.3–175.2)	108.7 ± 39.3 (83.7–133.7)	0.012 *	7.8, 50.3	0.58 (−0.26, 1.37)
Total sprint distance (m)	15.1 ± 7.5 (10.3–19.8)	12.1 ± 7.4 (7.5–16.8)	0.031 *	0.3, 5.6	0.40 (−0.42, 1.20)
Maximal speed (km·h^−1^)	29.2 ± 1.4 (28.3–30.1)	28.4 ± 1.4 (27.5–29.3)	0.031 *	0.1, 1.5	0.57 (−0.26, 1.37)
Body Load (au)	88.8 ± 28.0 (71.0–106.6)	100.5 ± 33.4 (79.2–121.7)	0.004 *	−18.7, −4.6	−0.38 (−1.17, 0.44)
**2nd Half**	**Official Matches**	**Friendly Matches**	** *p* **	**CI (95%)**	**Effect Size**
Duration (min)	43.4 ± 6.2 (39.4–47.3)	38.6 ± 4.2 (35.9–41.3)	0.016 *	1.1, 8.4	0.91 (0.04, 1.71)
Total Distance (m)	4531.9 ± 638.8 (4126.0–4937.8)	4002.6 ± 442.3 (3721.6–4283.6)	0.013 *	136.0, 922.7	0.96 (0.09, 1.77)
Average speed (m/min)	105.6 ± 14.3 (96.4–114.7)	104.9 ± 16.4 (94.5–115.3)	0.815	−5.3, 6.6	0.05 (−0.76, 0.84)
HSRD (m)	115.1 ± 33.9 (93.6–136.7)	111.9 ± 38.5 (87.4–136.4)	0.742	−17.9, 24.4	0.09 (−0.72, 0.89)
Total sprint distance (m)	15.4 ± 4.6 (12.5–18.4)	11.6 ± 6.5 (7.5–15.8)	0.145	−1.5, 9.2	0.67 (−0.17, 1.47)
Maximal speed (km·h^−1^)	28.9 ± 1.2 (28.1–29.7)	28.9 ± 1.1 (28.2–29.6)	0.974	−0.6, 0.7	0.00 (−0.80, 0.80)
Body Load (au)	75.1 ± 15.1 (65.5–84.7)	79.4 ± 32.1 (59.0–99.8)	0.520	−18.6, 10.0	−0.17 (−0.97, 0.64)

au, arbitrary units; m, meters; HSRD, high-speed running distance; AvS, average speed; TSD, total sprint distance; MS, maximal speed. * significant differences between official match vs. friendly match, *p* < 0.05.

**Table 3 healthcare-09-01708-t003:** Comparison of first and second halves data for official matches and friendly matches, respectively.

**Official Matches**	** *p* ** **(1st Half vs. 2nd Half)**	**Confidence Interval (95%)**	**Effect Size**
Duration (min)	0.034 *	0.3, 7.1	−1.38 (−2.22, −0.45)
Total Distance (m)	0.005 *	237.0, 1062.1	1.21 (0.30, 2.03)
Average speed (m/min)	0.079	−0.6, 10.0	0.38 (−0.44, 1.17)
HSRD (m)	0.057	−0.8, 46.0	0.47 (−0.36, 1.26)
Total sprint distance (m)	0.846	−4.4, 3.7	−0.05 (−0.85, 0.75)
Maximal speed (km·h^−1^)	0.322	−0.3, 1.0	0.23 (−0.58, 1.02)
Body Load (au)	0.021 *	2.5, 25.0	0.61 (−0.23, 1.41)
**Friendly Matches**	** *p* ** **(1st Half vs. 2nd Half)**	**Confidence Interval (95%)**	**Effect Size**
Duration (min)	<0.001 *	4.9, 12.3	2.44 (1.31, 3.39)
Total Distance (m)	<0.001 *	822.7, 1418.4	2.73 (1.54, 3.73)
Average speed (m/min)	0.377	−5.5, 13.4	0.29 (−0.52, 1.09)
HSRD (m)	0.622	−17.2, 10.8	−0.08 (−0.88, 0.72)
Total sprint distance (m)	0.851	−5.1, 6.1	0.07 (−0.73, 0.87)
Maximal speed (km·h^−1^)	0.316	−1.4, 0.5	−0.40 (−1.19, 0.42)
Body Load (au)	0.001 *	11.0, 31.1	0.64 (−0.20, 1.044)

au, arbitrary units; m, meters; HSRD, high-speed running distance; AvS, average speed; TSD, total sprint distance; MS, maximal speed. * denotes difference from 2nd half. all *p* < 0.05.

## Data Availability

Data are available through the corresponding authors: hadi.nobari1@gmail.com (Hadi Nobari) and rafaeloliveira@esdrm.ipsantarem.pt (Rafael Oliveira).

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
