# Peer review of "Variability of External Intensity Comparisons between Official and Friendly Soccer Matches in Professional Male Players"

_healthcare, 2021, doi:10.3390/healthcare9121708_

Round 1

Reviewer 1 Report

English must be improved.

L76 is misleading.

Introduction should be revised as it feels a sum of paragraphs with lack of connection.

Authors should be much more cautious interpreting their results; such a small sample leading to small ES.

It is not clear the relationship between FM in pre-season with FM during season.

Reviewer 2 Report

The aim of the study were to compare the external load between official and friendly matches, and between 1st and 2nd halves, in the Iranian Premier League.

The research is not innovative, nor does it bring any revealing elements to science, but it was carried out methodologically correctly.

The work is written following the steps of the scientific method.

The conclusions from the study may be more interesting. They should be written again in the points, giving specific results of own research, which clearly result from the tables.

The study is well designed. However I have some minor comments I’d like to express.

The Abstract must be improved, with a sequence of the following systematization: Objectives, Methods, Results, and Conclusions.

In the introduction, I propose to write a bit more about training loads, how to write and measure loads. It is worth paying more attention to the physiological reactions of the body to these loads.

Line 94. The matter is obvious. It is worth paying attention to what these differences between the first and second half may result from. be it the level of the match or the level of preparation of the players.

The aim of the work is clear whether a hypothesis is needed.

Line 100. How were the anthropometric measurements made, with what? Where?

It is worth adding a practical (appilcative) conclusion.

In my opinion, the conclusions should be more specific not generalized, but this is only a suggestion.

The article is generally valuable and correctly written, please treat the above comments only as suggestions. 

Reviewer 3 Report

According to the attached notes

Round 2

Reviewer 1 Report

Good job improving the quality of the manuscript.

Reviewer 2 Report

Accept in present form

This manuscript is a resubmission of an earlier submission. The following is a list of the peer review reports and author responses from that submission.